# Sand-Dust Image Enhancement Using Chromatic Variance Consistency and Gamma Correction-Based Dehazing

**DOI:** 10.3390/s22239048

**Published:** 2022-11-22

**Authors:** Jong-Ju Jeon, Tae-Hee Park, Il-Kyu Eom

**Affiliations:** 1Department of Electronics Engineering, Pusan National University, 2 Busandaehak-ro 63 Beon-gil, Busan 46241, Republic of Korea; 2Department of Mechatronics Engineering, Tongmyong University, 428 Sinseon-ro, Nam-gu, Busan 48520, Republic of Korea

**Keywords:** sand-dust image enhancement, chromatic variance consistency, dehazing, gamma correction, cross-correlation, color correction

## Abstract

In sand-dust environments, the low quality of images captured outdoors adversely affects many remote-based image processing and computer vision systems, because of severe color casts, low contrast, and poor visibility of sand-dust images. In such cases, conventional color correction methods do not guarantee appropriate performance in outdoor computer vision applications. In this paper, we present a novel color correction and dehazing algorithm for sand-dust image enhancement. First, we propose an effective color correction method that preserves the consistency of the chromatic variances and maintains the coincidence of the chromatic means. Next, a transmission map for image dehazing is estimated using the gamma correction for the enhancement of color-corrected sand-dust images. Finally, a cross-correlation-based chromatic histogram shift algorithm is proposed to reduce the reddish artifacts in the enhanced images. We performed extensive experiments for various sand-dust images and compared the performance of the proposed method to that of several existing state-of-the-art enhancement methods. The simulation results indicated that the proposed enhancement scheme outperforms the existing approaches in terms of both subjective and objective qualities.

## 1. Introduction

Outdoor images are captured in a wide range of light atmospheres. Due to uncontrollable factors, the performance of outdoor vision systems, such as intelligent surveillance systems, autonomous driving cars, image-based sensing systems, and traffic monitoring systems, can be degraded. In recent decades, sand-dust has been identified as a threat to human life and health. It also has an adverse effect on outdoor vision applications. In sand-dust weather conditions, captured images appear overall to be yellow or even red, because the radius of sand-dust particles is much larger than that of haze and fog particles [1]. Therefore, sand-dust images adversely affect various remote computer vision systems.

Figure 1 shows sand-dust image examples and their corresponding atmospheric light images and histograms. As shown in Figure 1, the acquired images appear to be overall yellow or even red because blue and green light is absorbed by sand-dust particles much more easily than red light. The aim of sand-dust image enhancement is real-time correction of these severe color casts and improvement of image contrast and detail. Since sand-dust images have different color distribution characteristics from hazy or underwater images, direct application of haze removal [2,3,4,5] or underwater image enhancement methods [6,7,8] to sand-dust images can lead to unsatisfactory results. Thus, sand-dust image enhancement is a more challenging problem than conventional dehazing or underwater image enhancement.

Recently, the amount of research on the enhancement of sand-dust images has been increasing. A physical image degradation model is frequently used for enhancing sand-dust images. The atmospheric scattering model (ASM) [9] is commonly used in sand-dust image restoration algorithms. Other popular approaches utilize various color correction algorithms customized to sand-dust weather conditions to enhance images. Machine learning-based enhancement approaches have not yet been actively pursued because there is no large benchmark dataset of sand-dust images. Various sand-dust image enhancement methods are briefly reviewed in the following section.

This paper presents a novel color correction-based sand-dust image enhancement method. The proposed method consists of three main states, namely initial color correction, enhancement, and final color correction.

The main contributions of this paper are as follows:Initial color correction—A novel color correction algorithm is introduced using the convex sum of the red or blue channels based on the green channel. The weight for the convex sum is obtained from the standard deviation (SD) of the color channels. We show that this method preserves the consistency of the variance of chromatic channels. In addition, we propose a background luminance-preserving image normalization method for maintaining the coincidence of the mean of each color channel;Enhancement—A dehazing framework is adopted to enhance color-corrected sand-dust images. The saturation-based transmission map estimation [4] is modified using the gamma correction technique. An image-adaptive gamma value is used to estimate the transmission map;Final color correction—We present a color pixel correction algorithm based on the cross-correlation of chromatic histograms to reduce reddish artifacts remaining in the enhanced images.The proposed method generates reasonable enhancement results in a wide variety of sand-dust images.

The remainder of this paper is organized as follows. Section 2 provides a brief review of sand-dust image enhancement methods. The motivation for this work is presented in Section 3. The proposed algorithm is presented in Section 4. In Section 5, the performance of the proposed method is compared to that of several existing methods using experimental results. Section 6 presents the conclusion.

## 2. Related Works

### 2.1. Image Degradation Model-Based Approaches

Image degradation model-based approaches mainly adopt the ASM, which is widely used for dehazing and underwater image enhancement. However, because the sand-dust images have a different color veil from the hazy or underwater images, the ASM is often applied with various color correction methods.

Huang et al. [10] presented a visibility restoration technique based on the dark channel prior (DCP) [2]. To prevent insufficient estimation of haze thickness, a haze thickness estimation module was proposed based on a Laplacian-based gamma correction, and the actual scene color was restored using the Laplacian-based white patch retinex technique. However, the algorithm produced a bluish artifact for most sand-dust images. The artifact appeared to exist because the weak blue channel and the strong red channel were not considered in the ASM. Yu et al. [11] proposed a sand-dust image enhancement method using the ASM and information loss constraint. Although this method accurately estimated the atmospheric light by iterating it to compensate for the color shift, the shortcoming was the emergence of halos. Shi et al. [12] presented a DCP-based image enhancement algorithm for halo elimination. The color components in the Lab color space were used to remove the color shift. Furthermore, the improved DCP was used to remove haze in the RGB color space. Finally, the brightness component in the Lab color space was stretched to enhance contrast. This method effectively improved the visibility of sand-dust images; however, the colors in recovered images were faint.

Sand-dust images have a weak blue component and a strong red component. Therefore, the conventional DCP approach cannot effectively remove sand-dust effects from images. Gao et al. [13] proposed a model-based sand-dust image restoration algorithm using the blue channel prior (BCP), which inverted the blue color component. The atmospheric light and the transmission map were estimated using the BCP. However, this method failed to recover degraded images with heavy sand-dust effects. Cheng et al. [14] employed blue channel compensation and used guided image filtering to enhance sand-dust images. The blue channel technology was used to improve image contrast and white balancing technology was exploited to resolve the color distortion. Multilayer decomposition and multiscale fusion technologies were used to restore important faded features and edge information. This approach generated reasonable enhancement results. However, the results for sand-dust images with severe color casts were unsatisfactory. Recently, Shi et al. [15] introduced an enhancement algorithm based on red and blue channels. To correct the color cast and enhance the visibility of a sand-dust image, the method proposed a red channel-based correction function for correcting color casting error and a blue channel-based dust particle removal for eliminating sand-dust particles. The method was fast; however, some restored images exhibited over-enhanced colors. Lee [16] presented a color balancing method using the singular value decomposition and an adjustable DCP to enhance color-balanced sand-dust images. However, this method did not completely remove dust veils and dimmed colors from the enhanced images.

### 2.2. Color Correction-Based Approaches

The main problem of sand-dust image enhancement is severe color shifts or casts. The general color correction-based algorithms [17,18] are not directly applicable to adjusting the color shifts or casting in sand-dust images. Therefore, color correction approaches adopted to sand-dust weather conditions are required to enhance sand-dust images.

Fu et al. [19] proposed a single image-based enhancement method for sand-dust images utilizing a fusion strategy. They generated two images with different brightness levels by applying two different gamma corrections to the same sand-dust image. The enhanced image was obtained by fusing the two gamma-corrected images with three weighted maps that represented sharpness, chromaticity, and prominence. The enhancement was reasonable in weak sand-dust conditions. However, the method failed to enhance images with heavy sand-dust effects. Yan et al. [20] adopted a global fuzzy enhancement [21] and band-limited histogram equalization for sand-dust image enhancement. In this method, a partially overlapped sub-block histogram equalization [22] was applied to further enhance image details. Although the contrast of restored sand-dust images was improved, the overall enhancement effect was unsatisfactory. Al-Ameen [23] introduced an enhancement algorithm based on tuned tri-threshold fuzzy intensification operators, which were applied to images with a different threshold for each color channel. However, the method did not fully eliminate color casts. Wang et al. [24] proposed a sand-dust image enhancement algorithm in the Lab color space. Two chromatic components were combined to perform color correction and saturation stretching. In addition, a fast local Laplacian filtering was employed to the lightness component to enhance image details. However, this method produced blueness in the enhanced images.

In recent years, Shi et al. [25] proposed a normalized gamma transformation-based contrast-limited adaptive histogram equalization for sand-dust image enhancement in the Lab color space. The contrast of the lightness component of the sand-dust images was enhanced using the contrast-limited adaptive histogram equalization [26], followed by a normalized gamma correction operation. For chromatic components, a gray-world-based color correction method was adopted. It generated reasonable sand-dust images. However, the performance was poor for severely degraded images, where it was unable to remove dust haze. Park et al. [27] presented an effective sand-dust image enhancement method using successive color balance with coincident chromatic histograms. This method introduced a pixel-adaptive color correction algorithm based on the SD of chromatic histograms, followed by a green-mean-preserving normalization technique. Furthermore, a histogram shifting method was introduced to make the red and blue histograms overlap with the green histogram as much as possible. It provided a reasonable restoring performance in terms of both subjective and objective measures. Gao et al. [28] proposed a color balance technique to restore sand-dust images in the Lab color space. In the first step, the lost value of the blue channel was compensated using the green channel. The brightness component in the Lab color space was decomposed by guided filtering to obtain the detail component. A nonlinear mapping function and a gamma function were also applied to the detail component to enhance the image detail information. This method is suitable for real-time applications and can also be applied to restore underwater and hazy images.

### 2.3. Machine Learning-Based Approaches

Machine learning-based techniques are widely applied in the field of image enhancement. However, machine learning-based methods have not yet been studied in earnest in the field of sand-dust image enhancement. Recently, a varicolored end-to-end dehazing network [29] was proposed to improve various types of varicolored hazy images. This network comprised a haze color correction module which generates a color balanced hazy image and a visibility improvement module to recover a haze-free image. Li et al. presented a semi-supervised varicolored dehazing network [30] based on a physically disentangled joint intra- and inter-domain adaptation paradigm. They exploited the knowledge of the ASM that estimated the atmospheric light to determine the color distortion degree. These two networks were useful against weak color veils; however, they did not effectively remove strong sand-dust veils. Lang et al. proposed an unsupervised learning algorithm [31] that used a histogram-based color correction image as an input to improve sand-dust images. This method generated acceptable image results; however, it often produced over-enhancement of faint colors in the restored images.

## 3. Motivation

Various color balance techniques, such as gray-world assumption (GWA) [32], shade-of-gray (SOG) [33], max-RGB [34], and gray-edge [17], can be used as an initial step in sand-dust image enhancement. Recently, Ancuti et al. [35] presented a color channel compensation (3C) method, which can be applied as a pre-processing or post-processing step of various image enhancement tasks. In addition to these methods, color correction algorithms specializing in improving sand-dust images have been proposed in combination with various image enhancement methods. However, almost all enhancement techniques do not perform successfully in cases of extreme color distortion, and over-enhancement frequently appears for strongly attenuated colors. In this paper, we investigate why the conventional color correction methods based on the mean of color channels frequently fail to adequately enhance sand-dust images.

Figure 2 presents the color correction results for a sample sand-dust image yielded by various compensation methods. As shown in Figure 2, the mean values of the red, green, and blue channels of the sample sand-dust image are 220, 155, and 71, respectively. These mean values represent a typical sand-dust image with a weak blue channel and a strong red channel. The GWA, SOG, and max-RGB methods tend to make the means of the red and blue color components similar to that of the green channel. However, the SD of the weak blue channel increases (from 22 to 49 for GWA, 45 for SOG, and 41 for max-RGB), which produces a bluish corrected image. For this reason, the corrected image has a bluish or slightly purple tint. The 3C method tends to change the blue and red histograms to match that of the green channel, but the SD value of the blue channel does not significantly increase. Because the corrected image exhibits narrow chromatic histograms, it may be dimmed.

The GWA, SOG, and max-RGB methods work on the same assumption that a different order of the Minkowski norm of the reflectance in a scene is achromatic. These color correction algorithms only consider the coincidence of chromatic means of the three channels, but not the SDs of the color channels. The 3C method is based on balancing two opponent color pairs toward the origin. All methods tend to equalize the mean of each color component. In most cases, during the sand-dust image color correction, the SD of the blue channel increases significantly when the weak blue channel is strengthened. This produces bluish or purplish images. Figure 3 presents color-compensated sand-dust image examples. As shown in Figure 3, the GWA, SOG, and max-RGB algorithms fail to recover severe color shifts in reddish sample images. The 3C method does not successfully enhance the sand-dust images, either.

In summary, we can conclude that the results of sand-dust image enhancement largely depend on the adopted color correction method. The existing color correction algorithms only consider the coincidence of the chromatic means of the three channels, but not the SDs of the color channels. In most cases, the SD of the blue channel increases significantly during the strengthening of a weak blue channel. In this paper, we introduce a novel color-correction based on chromatic SD consistency to enhance sand-dust images with a small color shift.

## 4. Proposed Method

### 4.1. Initial Color Correction

Let **I***_c_* (*c*∈{*r*,*g*,*b*}) be a color channel of image **I**, and let *I_c_*(**x**) be the pixel value of **I***_c_* at spatial location **x**. Bold characters represent matrices or color pixel vectors and non-bold italic characters imply scalar variables. In this paper, we introduce a convex sum method for compensating the color values based on the SD of a given color channel **I***_c_*. Let **I***_c_*_,*SD*_ be the first corrected color channel using the SD weight obtained by the following operation:(1)$$I_{c,SD}(x)=\alpha_{c}I_{c}(x)+(1-\alpha_{c})I_{g}(x)$$
where α*_c_* is a weight defined as follows:(2)$$\alpha_{c}=\frac{\sigma_{m}}{\sigma_{c}+\sigma_{g}}$$
where *σ_m_* = min(*σ_c_*,*σ_g_*). If *σ_m_* = *σ_c_* (*σ_g_* > *σ_c_*), it indicates that **I***_c_* is a weak channel, such as the blue channel in a sand-dust image. In this case, color channel **I***_c_* with a low SD will be significantly affected by the green channel. On the contrary, color channel **I***_c_* will not be strongly affected by the green channel if *σ_m_* = *σ_g_* (*σ_g_* < *σ_c_*). This situation can produce a strong red component in the sand-dust image.

We will now analyze the effect of the proposed color correction using the SD weight shown in (1). The variance of corrected channel **I***_c_*_,*SD*_ is calculated as follows:(3)$$\sigma_{c,SD}^{2}=\alpha_{c}^{2}\sigma_{c}^{2}+{\left(1-\alpha_{c}\right)}^{2}\sigma_{g}^{2}+2\alpha_{c}(1-\alpha_{c})\sigma_{c}\sigma_{g}\rho_{c,g},$$
where *σ_c_*_,1_ is the SD of **I***_c_*_,1_, and *ρ_c_*_,*g*_ is the correlation coefficient between channels **I***_c_* and **I***_g_*. When **I***_c_* is fully decayed or uniform, *σ_c_* = 0 and *α_c_* = 0. Thus, **I***_c_*_,*SD*_ equals **I***_g_*. On the other hand, when *σ_c_* = *σ_g_*, then, *α_c_* = 1/2. In this case, the variance of **I***_c_*_,*SD*_ (*σ_c_* = *σ_g_*) is simply calculated as follows:(4)$${\sigma_{c,SD}^{2}|}_{\sigma_{c}=\sigma_{g}}=\frac{1+\rho_{c,g}}{2}\sigma_{g}^{2}.$$

As can be seen, even if *σ_c_* = *σ_g_*, the variance of the corrected channel is not always the same as that of the green channel but rather depends on the correlation coefficient *ρ_c_*_,*g*_.

To further examine the effectiveness of the proposed color correction method, it is useful to consider (3) in terms of the SD ratio. Let *r_c_* be the SD ratio of the corrected channel to that of the green channel, that is the following:(5)$$r_{c}=\frac{\sigma_{c}}{\sigma_{g}}.$$

When *σ_g_* > *σ_c_*, we notice that *σ_m_ = σ_c_* and 0 < *r_c_* < 1. Substituting *σ_c_* = *r_c_σ_g_* and *σ_m_ = σ_c_* into (3), we obtain the following:(6)$$\sigma_{c,SD}^{2}=\frac{r_{c}^{4}+2\rho_{c,g}r_{c}^{2}+1}{{\left(1+r_{c}\right)}^{2}}\sigma_{g}^{2}.$$

In the proposed algorithm, the green channel is not changed in this step, thus, *σ_g_*_,*SD*_
*=σ_g_*. The SD ratio of the corrected channel, when *σ_g_* > *σ_c_*, *r_c_*_,*SD*_ (*σ_g_* > *σ_c_*), is calculated as follows:(7)$$r_{c,SD}(\sigma_{g}>\sigma_{c})=\frac{\sigma_{c,SD}}{\sigma_{g,SD}}=\frac{\sqrt{r_{c}^{4}+2\rho_{c,g}r_{c}^{2}+1}}{1+r_{c}}\;.$$

When *σ_g_* < *σ_c_*, *σ_m_* equals *σ_g_* and *r_c_* > 1. In a similar way, the SD ratio of the corrected color channels can be obtained as follows:(8)$$r_{c,SD}(\sigma_{g}<\sigma_{c})=\frac{\sigma_{c,SD}}{\sigma_{g,SD}}=\frac{\sqrt{2r_{c}^{2}(1+\rho_{c,g})}}{1+r_{c}}\;.$$

The term in the numeration in (7) can be rearranged to give the following:(9)$$r_{c}^{4}+2\rho_{c,g}r_{c}^{2}+1=r_{c}^{4}-2r_{c}^{2}+1+2\rho_{c,g}r_{c}^{2}+2r_{c}^{2}={\left(r_{c}^{2}-1\right)}^{2}+2r_{c}^{2}\left(1+\rho_{c,g}\right).$$

When *σ_g_* > *σ_c_*, 0 < *r_c_* < 1. Thus, the term (*r*^2^*_c_* − 1)^2^ in (9) is very small. Therefore, we can use the following approximation: (10)$$\frac{\sqrt{{\left(r_{c}^{2}-1\right)}^{2}+2r_{c}^{2}\left(1+\rho_{c,g}\right)}}{1+r_{c}}\approx \frac{\sqrt{2r_{c}^{2}\left(1+\rho_{c,g}\right)}}{1+r_{c}}.$$

From (10), we can notice that *r_c_*_,*SD*_(*σ_g_* > *σ_c_*) ≈ *r_c_*_,*SD*_(*σ_g_* < *σ_c_*). In conclusion, we can expect that the SD ratio of the compensated channel to the green channel is controlled by nearly the same condition regardless of illumination.

When *r_c_* is 0.5, it indicates that the SD of a certain color channel is half the SD of the green channel, that is *σ_c_* = 0.5*σ_g_*. From (7), the SD ratio of the compensated color channel becomes 0.833 when *ρ_c_*_,*g*_ = 1. The SD of this color channel changes to 0.833 of the SD of the green channel. Thus, the SD of a weak channel is increased using the proposed method. On the other hand, consider the case of *σ_c_* = 2*σ_g_*, that is, *r_c_* = 2. This case can represent correcting a strong red channel. Using (8), we obtain the value of *r_c_*_,*SD*_ = 1.333. The SD of a strong red channel, which initially was twice the SD of the green channel, is reduced to 1.333 of the SD of the green channel.

Park et al. [27] proposed a green-mean-preserving image normalization technique for sand-dust image enhancement based on the assumptions that all the color values are nearly equivalent and the color in each channel averages to gray over the entire image. This simple approach can effectively remove the color veil from sand-dust images. In this paper, we present a modified green-mean-preserving image normalization method based on the preservation of background luminance. The proposed normalization algorithm can prevent the color of the corrected image from fading.

Let *B*(**x**) be the background-luminance-preserving factor, defined as follows:(11)$$B(x)=\frac{I_{r}(x)+I_{g}(x)+I_{b}(x)}{I_{r,SD}(x)+I_{g,SD}(x)+I_{b,SD}(x)}.$$

Let *I_c_*_,*BLC*_(**x**) be the background luminance-compensated color-corrected channel, which is calculated as follows:(12)$$I_{c,BLC}(x)=I_{c,SD}(x)\;B(x).$$

If the background intensity of **I***_c_*_,*SD*_ is increased by the color correction of (1), *B*(**x**) is smaller than 1. Therefore, the background intensity of **I***_c_*_,*SD*_ remains the same as that of the input color channel **I***_c_*. In the opposite case, *B*(**x**) is greater than 1, and the background intensity is maintained. Let *I_c_*_,*LP*_(**x**) be the corrected color channel obtained using the background luminance-preserving image normalization, defined as follows:(13)$$I_{c,LP}(x)=\frac{I_{c,BLC}(x)-\mu \left(I_{c,BLC}\right)}{pct_{p}\left(I_{c,BLC}\right)-pct_{q}\left(I_{c,BLC}\right)}+\mu \left(I_{g,BLC}\right)\;,$$
where *μ*(**z**) is the mean of **z**, *pct_p_*(**z**) is the *p*-th percentile of **z**, and *q* = 100 − *p*. The percentile operation is used to remove outlier pixels from the corrected sand-dust image.

Figure 4 illustrates the results of the green-mean-preserving normalization and the proposed background-luminance-preserving image normalization methods. Each normalization algorithm is followed by the color correction technique of (1). As shown in Figure 4, the normalization results of a non-severe sand-dust image (the first column in Figure 4) are similar. However, the two normalization algorithms achieve different correction results for severe sand-dust image samples. The results using Park et al.’s [27] method produce grayish corrected images, where the red component is particularly strongly reduced. On the contrary, the proposed normalization technique generates more colorful correction images. From this comparison, it can be concluded that the background luminance preservation can effectively ensure the consistency of the mean of the color channels.

Finally, we exploit the well-known image adjustment algorithm to stretch the dynamic range of pixels, which results in image brightening. The last color corrected image, *I_c_*_,*CCF*_(**x**) is calculated by the following:(14)$$I_{c,CCF}(x)=\frac{I_{c,LP}(x)-\underset{c}{\min}\left(\underset{x}{\min}I_{c,LP}(x)\right)}{\underset{c}{\max}\left(\underset{x}{\max}I_{c,LP}(x)\right)-\underset{c}{\min}\left(\underset{x}{\min}I_{c,LP}(x)\right)}.$$

In this study, we discard color pixel values greater than 99% and smaller than 1%.

### 4.2. Enhancement Based on Dehazing Technique

Almost all of the sand-dust image enhancement methods use color correction and ASM-based dehazing techniques. In this paper, we adopt the saturation-based transmission estimation algorithm [4] and modify it using the gamma correction prior (GCP) [36] in the enhancement step to improve the quality of sand-dust images.

In the ASM [9], the degraded image is represented as follows:(15)$$H_{c}(x)=J_{c}(x)t(x)+A_{c}\left(1-t(x)\right),$$
where *H_c_*(**x**) is the degraded channel, *J_c_*(**x**) is the original color channel, *A_c_* is the atmospheric light estimated from **I***_c_*, and *t*(**x**) is the transmission map. If estimates of *t*(**x**) and *A_c_* are available, *J_c_*(**x**) can be recovered from a single image as follows:(16)$$J_{c}(x)=\frac{H_{c}(x)-A_{c}}{t(x)}+A_{c}.$$

In [4], the transmission map is calculated using the intensity and saturation components as follows:(17)$$t(x)=1-K_{H}(x)\left(1-\frac{S_{H}(x)}{S_{J}(x)}\right),$$
where *K***_H_**(**x**) is the intensity of the degraded image **H** normalized by **A**, and *S***_H_**(**x**) and *S***_J_**(**x**) are the saturation components of **H** and **J** in the HSI color space, respectively. Here, *t*(**x**) is a function of a single saturation value *S***_J_**(**x**). However, estimating *S***_J_**(**x**) is challenging. For a color pixel *G_c_*(**x**) at location **x** in the HSI color space, the intensity is defined as *K***_G_**(**x**) = (*G_r_*(**x**)+ *G_g_*(**x**)+ *G_b_*(**x**))/3, while the saturation is calculated as *S***_G_**(**x**) = 1 − *m***_G_**(**x**)/*K***_G_**(**x**), where *m***_G_**(**x**) is the minimum pixel value at location **x**. We modify *t*(**x**) in (17) using the definitions of intensity and saturation as follows:(18)$$t(x)=1-K_{H}(x)\left(1-\frac{1-\frac{m_{H}(x)}{K_{H}(x)}}{1-\frac{m_{J}(x)}{K_{J}(x)}}\right)=1-K_{H}(x)\frac{K_{J}(x)m_{H}(x)-K_{H}(x)m_{J}(x)}{K_{J}(x)K_{H}(x)-K_{H}(x)m_{J}(x)}.$$

In this study, two unknown values, *K***_J_**(**x**) and *m***_J_**(**x**), are estimated using the GCP. The GCP is introduced into the dehazing framework, and then used to estimate the transmission map. We simplify the GCP to the following form:(19)$$V_{c}(x)=H_{c}{\left(x\right)}^{\Gamma },$$
where *V_c_*(**x**) is the processed virtual pixel and Γ is the gamma correction factor. Here, *V_c_*(**x**) is used as an efficient initial image to estimate the transmission map and is successfully applied to enhance hazy images. Therefore, two unknown values *K***_J_**(**x**) and *m***_J_**(**x**) in (18) are estimated using the intensity and minimum values of **V** at location of **x** as follows:(20)$$K_{J}(x)\approx K_{V}(x),$$
(21)$$m_{J}(x)\approx m_{V}(x),$$

Finally, the transmission map is estimated as follows:(22)$$t(x)\approx 1-K_{H}(x)\frac{K_{V}(x)m_{H}(x)-K_{H}(x)m_{V}(x)}{K_{V}(x)K_{H}(x)-K_{H}(x)m_{V}(x)}.$$

Figure 5 shows a virtual image obtained using the simplified GCP for a sample hazy image with ground truth. As shown in Figure 5, haze is effectively removed from the obtained virtual image, which demonstrates a reasonable restoration of intensity and minimum values. However, it also exhibits poor restoration of color, manifested by dimmed color in the corrected images. The mean squared error (MSE) between the ground truth (**J**) and hazy image (**H**) is 9.93. The MSE between the ground truth (**J**) and gamma-corrected virtual image (**V**) is reduced to 4.71. Furthermore, the MSE between *K***_v_** and *K***_J_** is 3.45, and the MSE between *m***_v_** and *m***_J_** is 2.32. In conclusion, a virtual image **V** can effectively be used as an initial image to estimate the transmission map.

To obtain **V** using (19), the value of Γ is determined. Large Γ values are typically suitable for images with heavy haze (we actually use normalized pixel values between 0 and 1), while small Γ values can be used for of lightly hazed images. In this paper, we present a simple method for determining an image-adaptive Γ value to obtain a virtual image. The image-adaptive Γ value for each hazy image is determined as a simple exponentially increasing function of the image mean value with the maximum (Γ_max_) and minimum (Γ_min_ = 1) limits, as follows:(23)$$\Gamma =pe^{\mu (H)}+q.$$
where *p* and *q* are the coefficients that determine the shape of the exponential function, which are easily calculated using two conditions of (*μ*(**H**),Γ) = (0,Γ_min_) and (*μ*(**H**),Γ) = (1,Γ_max_).

In the ASM of (15), the enhanced image is obtained by replacing **H***_c_* with **I***_c_*_,*CCF*_ of (14) to calculate *A_c_*_,*CCF*_, *t_CCF_*(**x**), and Γ*_CCF_*. We can, thus, produce an enhanced image by using **I***_c_*_,*CCF*_ in (16), as follows:(24)$$I_{c,E}(x)=\frac{I_{c,CCF}(x)-A_{c,CCF}}{t_{CCF}(x)}+A_{c,CCF},$$
where *I_c_*_,*E*_(**x**) is the enhanced sand-dust image.

Figure 6 shows the enhanced images produced by applying (22) with *μ*(**I***_c_*_,*CCF*_) and the corresponding Γ*_CCF_* value when Γ_max_ is assumed as 2. As shown in Figure 6, **I***_c_*_,*CCF*_ images are enhanced with different Γ*_CCF_* values. As the value of Γ_max_ increases, a stronger enhancement is achieved. However, a strong enhancement can cause the red pixels to stand out. To prevent this, we introduce a simple color compensation algorithm using cross-correlation to achieve the coincidence of chromatic histograms.

### 4.3. Final Color Correction

The enhancement process increases the image contrast and colors. As shown in the sample images at the bottom of Figure 6, some color components, especially red, can be strengthened through the enhancement process. It is known that the overlap area of the color histograms reflects the variation in the color temperature of the image, and the maximum overlap area is responsible for the standard color temperature. Therefore, histogram shifting algorithms [27,37] make the red and blue histograms overlap the green histogram as much as possible. However, the shifting methods based on the maximum overlap area have a limitation of the shifting range because the computational cost increases rapidly as the shifting range increases. In this paper, we introduce a novel chromatic histogram coincidence method using the cross-correlation technique with no restriction on the range of shifting with the same computational cost.

Before performing the cross-correlation operation, the green-mean-preserving normalization method [27] is applied to the enhanced image *I_c_*_,*E*_(**x**), which generates *I*′*_c_*_,*E*_(**x**). To avoid the effect of background luminance, the background-normalized pixel value for *I*′*_c_*_,*E*_(**x**) is obtained as follows:(25)$$I_{c,E}^{n}(x)=\frac{I_{c,E}^{\prime }(x)}{I_{r,E}^{\prime }(x)+I_{g,E}^{\prime }(x)+I_{b,E}^{\prime }(x)},$$
where *I^n^_c_*_,*E*_(**x**) is the normalized pixel value. Let **h***_c_*_,*E*_ be the histogram function of *I^n^_c_*_,*E*_(**x**) with 256 bins and let *h_c_*_,*E*_(*k*) be the number of pixels with a value of *k* (0 ≤ *k* ≤ 255). The cross-correlation between the green histogram, **h***_g_*_,*E*_, and the other color histogram, **h***_c_*_,*E*_, is defined as follows:(26)$$A_{gc}(\tau_{c})=\sum_{k}h_{g,E}(k)h_{c,E}(k+\tau_{c}),$$
where *A_gc_*(*τ_c_*) is the cross-correlation, and *τ_c_* (−255 ≤ *τ_c_* ≤ 255) is the lag parameter. We assume that the coincidence of the chromatic histograms occurs when the *A_gc_*(*τ_c_*) value reaches its maximum. The maximum lag *τ_c_*_,max_ is obtained as follows:(27)$$\tau_{c,\max}=\underset{\tau_{c}}{\arg\max}A_{gc}(\tau_{c}).$$

Using *τ_c_*_,max_, the final color corrected image is computed as follows:(28)$$I_{c,CCL}(x)=I_{c,E}^{\prime }(x)+\tau_{c,\max},$$
where *I_c_*_,*CCL*_(**x**) is the final enhanced sand-dust image. According to the definition of cross-correlation, *τ_g_*_,max_ is always 0, which results in no shift in the green histogram

Figure 7 shows several examples of the resultant enhanced image samples using the final color correction algorithm. In the corrected image with a light sand-dust effect in the first column from the left, the maximum lag values corresponding to the red and blue histograms are negligible. The enhanced image in the second column has *τ_r_*_,max_ = −15 and *τ_r_*_,max_ = 1, which results in a reduced red component, indicating a more robust image enhancement. In the last column, an enhanced image with a slightly reduced red and slightly strengthened blue channel is demonstrated. In conclusion, the cross-correlation-based final color correction strategy can adaptively shift the chromatic histograms and fine-tune the color cast.

### 4.4. Summary of Proposed Method

Figure 8 shows the steps in the entire algorithm of the proposed sand-dust image enhancement method. For a given sand-dust image **I***_c_*, the first corrected color channel **I***_c_*,*_SD_* is obtained based on the SD weight using (1). The initial color correction is performed, using (13) and (14) in a row. In the enhancement step, the atmospheric light, *A_c_*_,*CCF*_, is estimated using Tang et al.’s [38] method. We essentially use (17) to estimate the transmission map, *t_CCF_*. It is known that dehazing results are unaffected by different *A_c_*_,*CCF*_ estimation methods [4]. The enhanced image **I***_c_*_,*E*_ is computed using (24). The restored image **I***_c_*_,*CCL*_ undergoes final color correction strategy and is obtained using *τ_c_*_,max_ and the pixel sift algorithm based on (28).

## 5. Simulation Results

To verify the effectiveness of the proposed sand-dust image enhancement method, we test it on 245 sand-dust images [27]. The performance of the proposed approach is compared to those of five state-of-the-art algorithms, namely, fusion-based enhancement (FBE) [19], visibility enhancement via tri-threshold fuzzy intensification operators (TFIO) [23], halo-reduced dark channel prior (HRDCP) [12], normalized gamma transformation (NGT) [25], and successive color balance (SCB) [27]. The codes for the five sand-dust image enhancement methods were downloaded from their respective project sites, and the code of the proposed algorithm and sand-dust image dataset are available in [39].

### 5.1. Computation Time

The proposed method was implemented on an Intel i7–107000K CPU @ 3.80 GHz processor with 48 GB RAM without multithreading acceleration. The code was written in non-optimized Python in the Linux Ubuntu 18.04.6 LTS environment. The comparison methods were implemented in the same environment. The execution time was averaged over 10 runs. The durations of image reading and writing were excluded.

Table 1 lists the execution times for various image sizes. As shown in Table 1, on the average, FBE, TFIO, and HRDCP require long execution times, while SCB and NGT execute the required operations faster than the remaining methods. The execution times of our algorithm are generally short for small images and achieve the third rank for large images.

### 5.2. Ablation Study

The proposed method comprises three steps. We first used various quantitative measures to evaluate the effectiveness of the proposed method by examining the results of each step. Because there are no commonly agreed-upon quantitative evaluation measures for sand-dust images, we adopted the following four popular non-reference quality measures to examine the approximate tendency of quantitative image quality achievement: the naturalness image quality evaluator (NIQE) [40], no-reference perception-based image quality evaluator (PIQE) [41], blind/referenceless image spatial quality evaluator (BRISQUE) [42], and no-reference image quality metric for contrast distortion (NIQMC) [43]. Smaller NIQE, PIQUE, and BRISQUE, values represent better image qualities, whereas, for NIQMC, the reverse is true.

Table 2 presents average quantitative metric values according to the progress of each step. As the step progresses, the NIQMC score gets better, while the PIQE score gets worse. On the other hand, two quantitative values, including NIQE and BRISQUE, exhibit the highest score when only Step 1 (**I***_c_*_,*CCF*_) is applied. These two metric values degrade slightly in Step 2 (**I***_c_*_,*E*_) and then achieve better scores in the final step (**I***_c_*_,*CCL*_).

Figure 9 illustrates the intermediate results of the presented enhancement process. The SD weight-based correction algorithm is designed to obtain the chromatic variance consistency. Therefore, the intermediate image **I***_c_*_,*SD*_ has not yet removed the sand-dust veil, as shown in the second row of Figure 9. The sand-dust veil is removed after applying the proposed background-luminance-preserving image normalization algorithm. The obtained image **I***_c_*_,*CCF*_ can be used as an enhanced result when the hazy effect on the sand-dust image is weak. To eliminate the hazy effect, we introduce the dehazing framework. The result of the fourth row of Figure 9 shows the result **I***_c_*_,*E*_ of performing haze removal on **I***_c_*_,*CCF*_. Because the dehazing is basically accompanied by image sharpening, color information can be distorted. Therefore, the proposed method finally uses the pixel shifting method to remove color distortion caused by the haze removal. The fifth row of Figure 9 presents the final corrected result **I***_c_*_,*CCL*_. The result images are enhanced, and slight color casts are restored. From the results of the ablation study, it is believed that the quantitative metric values for enhanced sand-dust images do not yet have great reliability. Therefore, we compare our method with the existing methods based on the subjective image quality even though the quantitative metric values are reduced.

### 5.3. Qualitative Comparison

We qualitatively compare the proposed enhancement method to the five conventional methods. Figure 10 shows a qualitative comparison of our results on sand-dust images with letters added after capturing with those of the five existing methods. As shown in Figure 10, FBE removes a reddish color veil; however, color shifts occur and the color of the letters is not restored (white letters change to bluish). The letters in the images enhanced by TFIO are unchanged because this algorithm fails to remove the color veils from the sand-dust images. The HRDCP method does not effectively enhance the sand-dust images. Although the restored images obtained using this method have a high contrast, they show color cast and dimmed colors, and look unnatural. In addition, the restored letters exhibit a severe bluish artifact. Here, NGT effectively removes sand-dust veils. However, the enhanced images have a low-contrast, dimmed colors, and bluish letters. The SCB approach produces reasonable qualitative results with a small color cast, high contrast, and good brightness. However, this algorithm fails to fully restore the color of letters in the images. In comparison, the proposed method produces the best qualitative results with high contrast, good brightness, and nearly perfect restoration of letters in the images.

Because there are no ground truths for the sand-dust images used in the comparison, the letters or logs added after image capturing can serve as a reference for comparing the performance of various enhancement methods. From this point of view, it can be seen that the proposed sand-dust image enhancement scheme is the best among the compared methods.

The enhancement results of sand-dust images with a considerable number of color veils are presented in Figure 11. The FBE method occasionally fails to restore sand-dust images and, thus, its image enhancement performance cannot always be guaranteed. The TFIO and HRDCP methods do not remove sand-dust veils and generate poorly enhanced images. The NGT method produces enhanced images with dimmed colors and does not fully restore degraded images. The SCB approach provides reasonable restoration results. Nevertheless, small color distortions and over-enhanced color components are noticeable. However, the restored images obtained using the proposed method do not show color casts and have an acceptable contrast and brightness comparable to the SBC.

Figure 12 shows the enhancement results for almost fully reddish sand-dust images, referred to as “red-storm” images. The FBE, TFIO, and HRDCP methods do not restore the red-storm images at all, and these produce unnatural images. The NGT restoration is insufficient. SCB does not fully remove reddish veils and occasionally produces bluish artifacts. The proposed method does not restore red-storm images perfectly; however, it enhances the degraded images to an acceptable level. Because the enhancement of a red-storm image is a very difficult task, the proposed algorithm can be considered to have performed acceptably.

### 5.4. Quantitative Comparison

Table 3 shows a comparison of the four average quantitative metric values for all 245 test images. The proposed algorithm has the best average NIQE score. The SCB method has the second best NIQE score, and TFIO achieves a good NIQE score despite severe color distortion. The FBE, HRDCP, and NGT methods achieve NIQE scores similar to the subjective image qualities shown in Figure 10, Figure 11 and Figure 12. The SCB method achieves the best average PIQE and BRISQUE score values, followed by the proposed method. As shown in Table 3, the highest NIQMC scores are obtained by the proposed algorithm and FBE. The proposed method and SCB show the best performance when considering the scores of four quantitative measures.

## 6. Conclusions

Typical color correction methods aim at achieving chromatic mean consistency. However, achieving the mean consistency and guaranteeing good image enhancement performance when there are severe color distortions, such as those in sand-dust images, is challenging. In this paper, we proposed a novel color correction method that can achieve both the consistency of the chromatic variances and the coincidence of the chromatic means. We also mathematically demonstrated that the SD-based weighted convex sum method can match the variance of the chromatic components. The dehazing-based enhancement process using the image-adaptive gamma correction prior for estimating the transmission map was presented. Finally, the cross-correlation-based chromatic histogram shift algorithm was proposed to eliminate the remaining small color distortion from the enhanced images. The performance of the proposed method was compared to that of several existing sand-dust image enhancement algorithms. Experiments demonstrated that the proposed color correction algorithm was suitable for enhancing sand-dust images with severe color distortion.

## Figures and Tables

**Figure 1 sensors-22-09048-f001:**
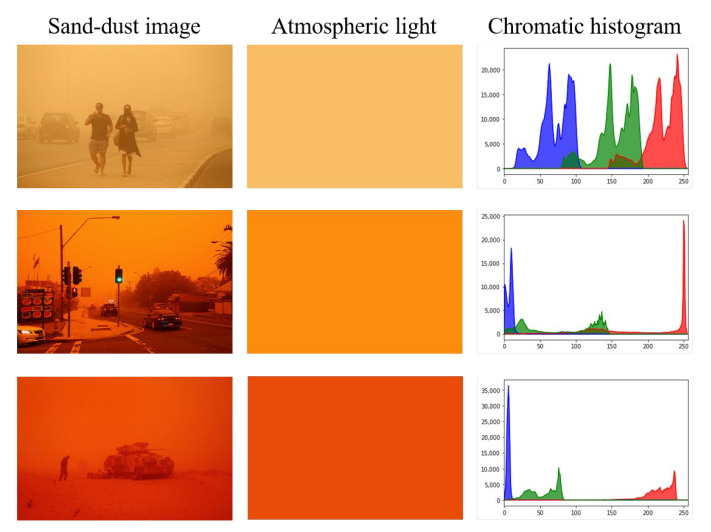
Sand-dust image and corresponding atmospheric light images and chromatic histograms.

**Figure 2 sensors-22-09048-f002:**
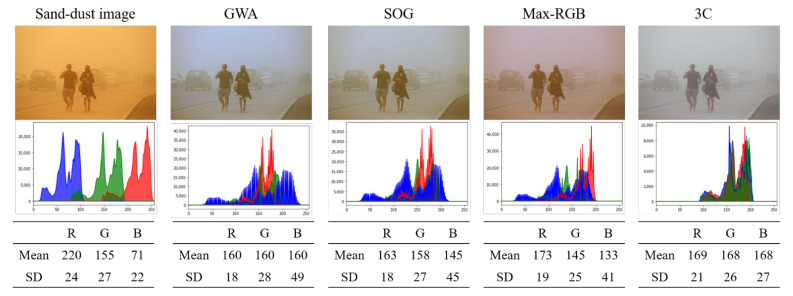
Color correction results yielded by various color compensation methods.

**Figure 3 sensors-22-09048-f003:**
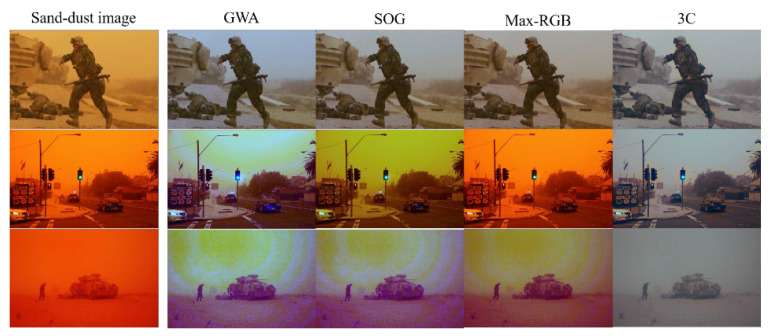
Sand-dust image enhancement results using color correction followed by an image enhancement strategy.

**Figure 4 sensors-22-09048-f004:**
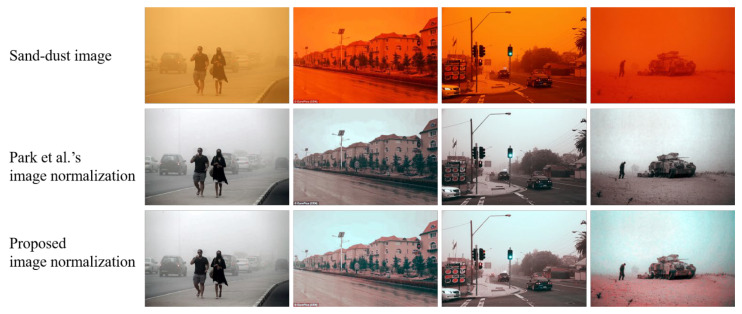
Image normalization comparison between Park et al.’s image normalization [27] and the proposed background-luminance-preserving method.

**Figure 5 sensors-22-09048-f005:**
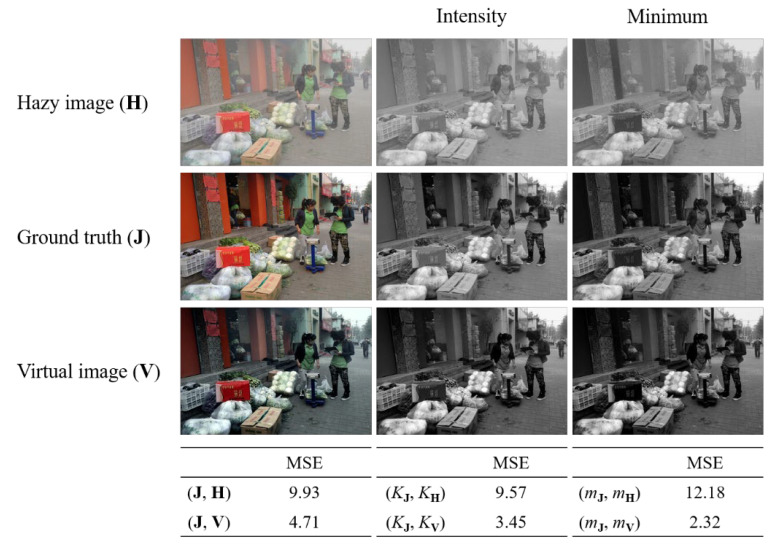
Virtual image using GCP for a sample hazy image with ground truth. Intensity and minimum images for three images and mean squared error values corresponding to hazy and virtual images for ground truth are drawn.

**Figure 6 sensors-22-09048-f006:**
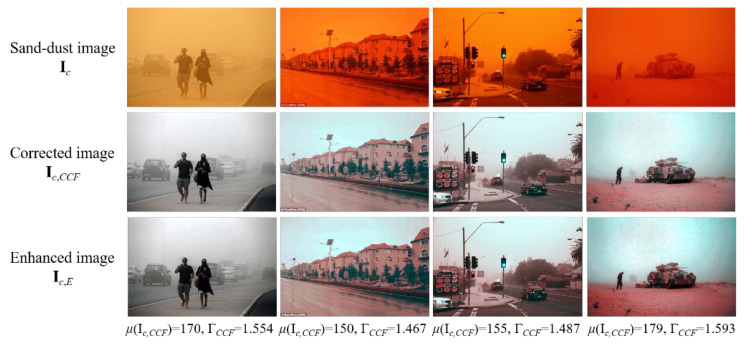
Enhanced images produced using the proposed dehazing method. Mean values of color- corrected images and corresponding gamma values are shown for Γ_max_ = 2.

**Figure 7 sensors-22-09048-f007:**
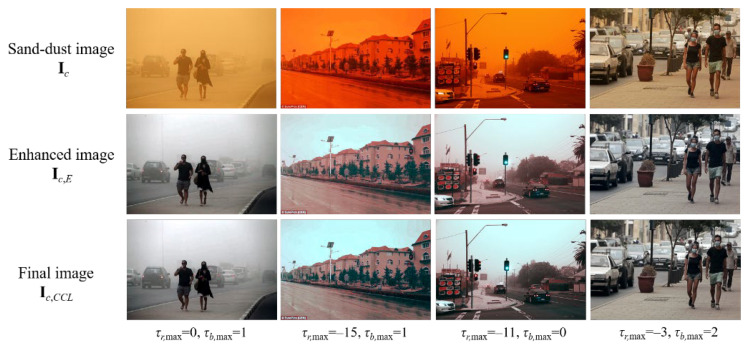
Fine-tuned images using a cross-correlation-based chromatic histogram shift.

**Figure 8 sensors-22-09048-f008:**
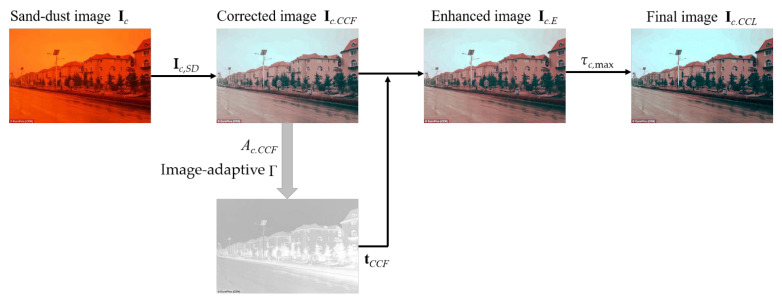
Overall algorithm of proposed method.

**Figure 9 sensors-22-09048-f009:**
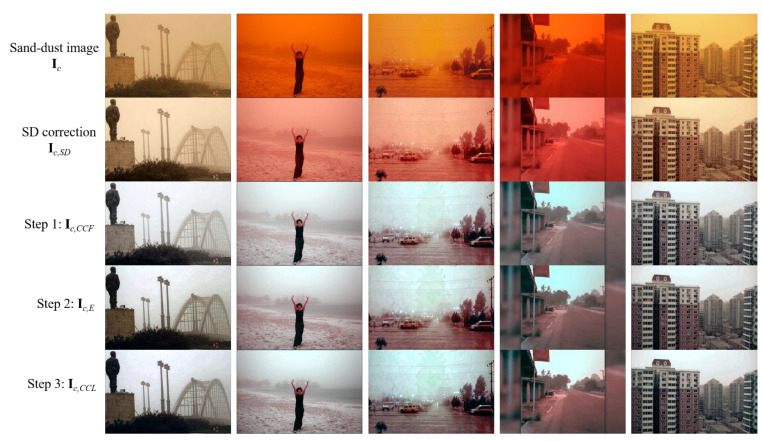
Intermediate results of proposed sand-dust image enhancement process.

**Figure 10 sensors-22-09048-f010:**
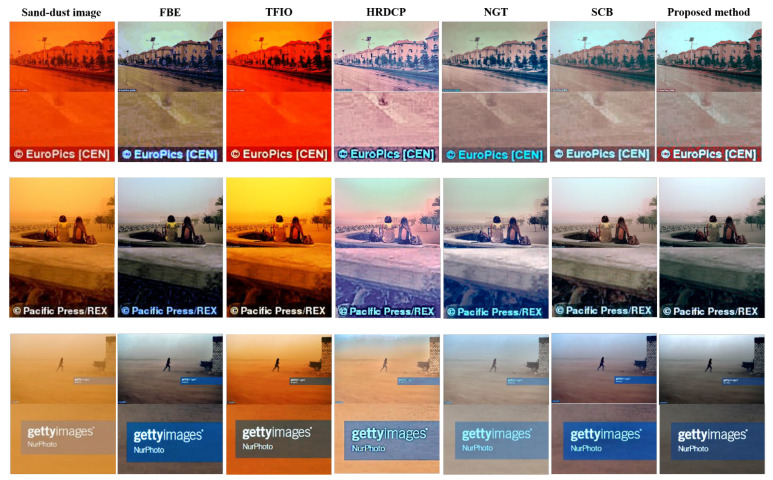
Qualitative comparison of enhanced sand-dust images with letters added after capturing.

**Figure 11 sensors-22-09048-f011:**
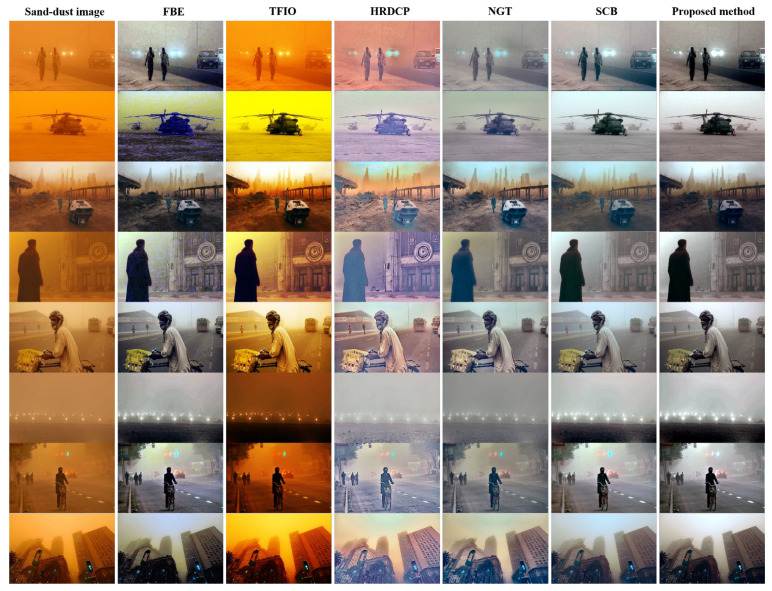
Enhancement results of various sand-dust images with color veils.

**Figure 12 sensors-22-09048-f012:**
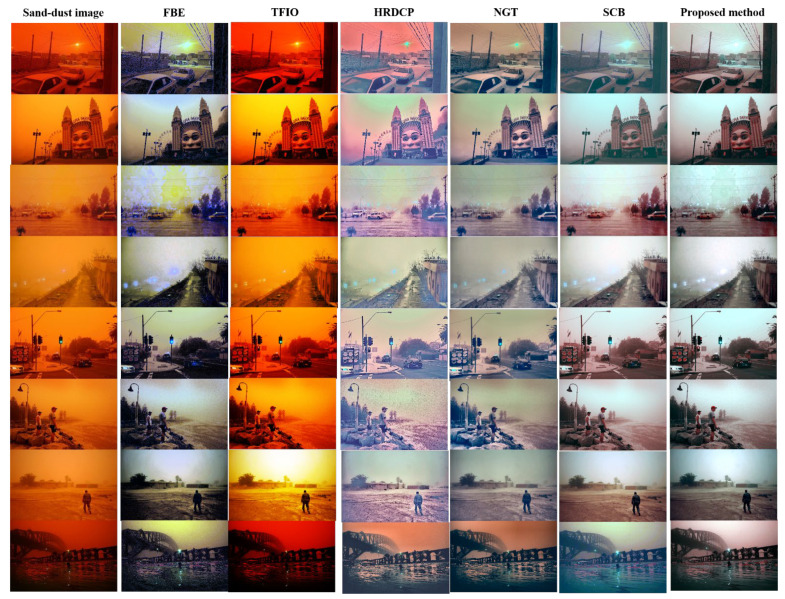
Enhancement results of various red-storm images with heavy reddish veils.

**Table 1 sensors-22-09048-t001:** Execution times of various sand-dust image enhancement methods (Unit—second).

Image Size				
Method	640 × 360	1000 × 750	2000 × 1091	4032 × 3024
FBE [19]	0.160	0.644	2.187	9.856
TFIO [23]	0.034	0.221	1.834	28.807
HRDCP [12]	1.095	4.306	14.128	64.767
NGT [25]	0.048	0.192	0.582	2.696
SCB [27]	0.214	0.232	0.443	1.433
Proposed method	0.051	0.233	0.775	3.828

**Table 2 sensors-22-09048-t002:** Average quantitative metric values for all test images according to the progress of each step.

Step	NIQE	PIQE	BRISQUE	NIQMC
Step 1: **I***_c_*_,*CCF*_	3.378	42.554	31.768	5.415
Step 2: **I***_c_*_,*E*_	3.417	43.776	32.202	5.483
Step 3: **I***_c_*_,*CCL*_	3.404	44.155	32.141	5.537

**Table 3 sensors-22-09048-t003:** Average quantitative metric values for all test images.

Method	NIQE	PIQE	BRISQUE	NIQMC
FBE [19]	3.741	47.854	34.747	5.450
TFIO [23]	3.474	46.255	36.538	5.071
HRDCP [12]	4.123	46.111	32.034	4.786
NGT [25]	3.525	47.582	34.565	4.626
SCB [27]	3.472	43.938	29.465	5.379
Proposed method	3.404	44.155	32.141	5.537

## Data Availability

We used a 245 sand-dust test image dataset, which is available at https://sites.google.com/view/ispl-pnu/, accessed on 10 October 2022.

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
