# Peer review of "Sand-Dust Image Enhancement Using Chromatic Variance Consistency and Gamma Correction-Based Dehazing"

_sensors, 2022, doi:10.3390/s22239048_

Round 1

Reviewer 1 Report

In this work, the authors propose a three-stage image enhancement framework including color correction, dehazing, and artifact removal for sand-dust image enhancement. The idea is intuitive and the experimental results are good. I have several suggestions to further improve the manuscript.

1. The introduction of the related works about varicolored haze removal [1, 2] is missing. The difference analysis from these highly-related previous works is required.

[1] Varicolored image de-hazing. CVPR, 2020.

[2] Physically disentangled intra- and inter-domain adaptation for varicolored haze removal. CVPR, 2022.

2. The ablation of each step is missing. It is not clear how each stage works. Both the qualitative and quantitative results are suggested for better illustration.   

3. Each step in the proposed method seems to borrow from the existing work. Please clarify the difference and motivation more clearly. 

Author Response

  1. The introduction of the related works about varicolored haze removal [1, 2] is missing. The difference analysis from these highly-related previous works is required.

[1] Varicolored image de-hazing. CVPR, 2020.

[2] Physically disentangled intra- and inter-domain adaptation for varicolored haze removal. CVPR, 2022.

Answer: I missed these papers because I searched for papers with the keyword ‘sand-dust image’. I added one more paper published in 2022. We changed some sentences in the introduction to reflect reviewer’s comment. Detailed descriptions are introduced in the newly inserted a sub-section 2.3.

  1. The ablation of each step is missing. It is not clear how each stage works. Both the qualitative and quantitative results are suggested for better illustration.   

Answer: We inserted a sub-section 5.2 for ablation study.

  1. Each step in the proposed method seems to borrow from the existing work. Please clarify the difference and motivation more clearly. 

Answer: The proposed method comprises three steps. In the initial color correction step, we propose a novel correction algorithm using the SD-based weight. In addition, we propose a background luminance-preserving image normalization method for maintaining the coincidence of the mean of each color channel. This is a modified green-mean-preserving image normalization method [27] based on the preservation of background luminance. The effect of the proposed algorithm is illustrated in Figure 4. We clarified the difference between the proposed method and [27] in the revised paper.

In the second step, the gamma correction-based transmission map estimation is proposed. 

In the final step, we used a cross-correlating-based pixel shift method. The conventional methods used the maximum overlap area for pixel shift. This shift algorithm has a limitation of the shifting range because the computational cost increases rapidly as the shifting range increases. The proposed algorithm has no restriction on the range of shifting with the same computational cost.

The detailed descriptions are presented in the highlighted sentences in the revised paper.

Reviewer 2 Report

This paper presents a sand-dust image enhancement using chromatic variance consistency and Gamma correction-based dehazing. This research contents of this paper are of great significance. And interesting results have been demonstrated by authors. Please make some minor revisions to achieve final publication standard.

1. Table 1 shows workflow of the proposed algorithm. I suggest authors change it as the format of Pseudo chart.

2. Please indicate the difference image enhancement in sand hust scenarios and that in general vague environment.

3. Please indicate application range of the proposal in manuscript. Is it suitable specifically for sand hust scenarios, or also some other scenarios?

4. In comparison experiments, it is suggested to renamed the proposal. Currently, it is named as "Proposed". It is not usual.

5. Please pay attention to some new research works in related area, such as:

1) DeformableGAN: Generating Medical Images With Improved Integrity for Healthcare Cyber Physical Systems, doi: 10.1109/TNSE.2022.3190765.

2) Bl-IEA: A Bit-level Image Encryption Algorithm for Cognitive Services in Intelligent Transportation Systems, doi: 10.1109/TITS.2021.3129598.

Author Response

  1. Table 1 shows workflow of the proposed algorithm. I suggest authors change it as the format of Pseudo chart.

    Answer: We changed Table 1 to pseudo chart using images (Figure 8), and also changed sentences to describe Figure 8.

    2. Please indicate the difference image enhancement in sand dust scenarios and that in general vague environment.

    Answer: The requested descriptions are presented in Introduction part as follows: The aim of sand-dust image enhancement is real-time correction of these severe color casts and improvement of image contrast and detail. Since sand-dust images have different color distribution characteristics from hazy or underwater images, direct application of haze removal or underwater image enhancement methods [6-8] to sand-dust images can lead to unsatisfactory results. Thus, sand-dust image enhancement is a more challenging problem than conventional dehazing or underwater image enhancement.

    3. Please indicate application range of the proposal in manuscript. Is it suitable specifically for sand dust scenarios, or also some other scenarios?

    Answer: The proposed algorithm of this paper was designed specifically for enhancing sand-dust images. The title of this paper includes ‘Sand-dust Image Enhancement’.

    4. In comparison experiments, it is suggested to renamed the proposal. Currently, it is named as "Proposed". It is not usual.

    Answer: We used the term ‘Proposed method’ for tables and figures.

    5. Please pay attention to some new research works in related area, such as:

    1) DeformableGAN: Generating Medical Images With Improved Integrity for Healthcare Cyber Physical Systems, doi: 10.1109/TNSE.2022.3190765.

    2) Bl-IEA: A Bit-level Image Encryption Algorithm for Cognitive Services in Intelligent Transportation Systems, doi: 10.1109/TITS.2021.3129598.

    Answer: Paper 1) presented a method for GANs to generate images without producing checkerboard artifacts. Paper 2) was not presented for image enhancement, but is for image encryption. However, I think these two papers are not directly related to sand-dust image enhancement.

Reviewer 3 Report

In sand-dust environments, the low-quality of images captured outdoors affects adversely many remote-based image processing and computer vision systems, because of severe color casts, low contrast, and poor visibility of sand-dust images. This manuscript is attractive and useful, but it slao have shortcomings

1. In "Related work", the deep learning-based methods are also need be introduced.

2. The resolution of Figure.1 is disillusionary, and it need be bigger than now, which can make it clear to the reader.

3. In experiment, the deep learning-based methods are also used as comparison.

Author Response

  1. In "Related work", the deep learning-based methods are also need be introduced.

Answer: Detailed descriptions for the deep learning-based methods are introduced in the newly inserted sub-section 2.3.

2. The resolution of Figure.1 is disillusionary, and it need be bigger than now, which can make it clear to the reader.

Answer: We replaced Figure 1 with an image with a higher resolution, and enlarged the size of the figure.

3. In experiment, the deep learning-based methods are also used as comparison.

Answer: We found three recently published deep learning-based methods for sand-dust image enhancement. Two methods did not provide their codes. One method provided a site with code, but the code has not yet been uploaded on the site. Therefore, we could not use machine learning-based methods for comparison. I hope your generous mind.

Round 2

Reviewer 2 Report

It can be accepted.

Reviewer 3 Report

  • I think this version can be accepted.